# Shorter Survival after Liver Pedicle Clamping in Patients Undergoing Liver Resection for Hepatocellular Carcinoma Revealed by a Systematic Review and Meta-Analysis

**DOI:** 10.3390/cancers13040637

**Published:** 2021-02-05

**Authors:** Charles-Henri Wassmer, Beat Moeckli, Thierry Berney, Christian Toso, Lorenzo A. Orci

**Affiliations:** Division of Abdominal and Transplantation Surgery, Department of Surgery, Faculty of Medicine, Geneva University Hospitals, 4 rue Gabrielle-Perret-Gentil, 1205 Geneva, Switzerland; thierry.berney@hcuge.ch (T.B.); christian.toso@hcuge.ch (C.T.); lorenzo.orci@hcuge.ch (L.A.O.)

**Keywords:** Pringle, hepatocellular carcinoma, liver resection, portal clamping, survival

## Abstract

**Simple Summary:**

Hepatocellular carcinoma (HCC) is the most prevalent tumor of the liver and represents the second most common cause of oncological-related deaths worldwide. Despite all progress made in the field, surgical resection or liver transplantation are, at the moment, the only curative therapies available. Liver resection, especially for large, central tumors, are at risk of important bleeding. Significative hemorrhage during HCC resections have been linked to an increased rate of post-operative complications and tumor recurrence. Therefore, hepatic pedicle clamping during surgery has been used in order to reduce the bleeding risks. However, this method induces ischemia/reperfusion injuries, which has also been associated with tumor recurrence. For this reason, we aimed to evaluate if pedicle clamping is indeed associated with tumor recurrence and shorter survival, by performing a systematic review of the literature and meta-analysis.

**Abstract:**

Liver pedicle clamping minimizes surgical bleeding during hepatectomy. However, by inducing ischemia-reperfusion injury to the remnant liver, pedicle clamping may be associated with tumor recurrence in the regenerating liver. Hepatocellular carcinoma (HCC) having a high rate of recurrence, evidences demonstrating an eventual association with pedicle clamping is strongly needed. We did a systematic review of the literature until April 2020, looking at studies reporting the impact of liver pedicle clamping on long-term outcomes in patients undergoing liver resection for HCC. Primary and secondary outcomes were overall survival (OS) and disease-free survival, respectively. Results were obtained by random-effect meta-analysis and expressed as standardized mean difference (SMD). Eleven studies were included, accounting for 8087 patients. Results of seven studies were pooled in a meta-analysis. Findings indicated that, as compared to control patients who did not receive liver pedicle clamping, those who did had a significantly shorter OS (SMD = −0.172, 95%CI: −0.298 to −0.047, *p* = 0.007, *I*^2^ = 76.8%) and higher tumor recurrence rates (odds ratio 1.36 1.01 to 1.83. *p* = 0.044, *I*^2^ = 50.7%). This meta-analysis suggests that liver pedicle clamping may have a deleterious impact on long-term outcomes. An individual patient-data meta-analysis of randomized trials evaluating liver pedicle clamping is urgently needed.

## 1. Introduction

Hepatocellular carcinoma (HCC) is the most common primary malignancy of the liver. It is a leading cause for cancer-related death at the global stage, accounting for nearly 745,000 deaths per year [1,2]. Due to late diagnosis and limited therapeutic options, HCC has an elevated mortality rate, with an overall five-year survival rate below 20% [3]. Among several therapeutic approaches [4], hepatectomy and liver transplantation are the only curative options. Liver resection in particular, is indicated in patients with early-stage HCC and preserved liver function [5].

In patients with large tumors or those in anatomically difficult locations, liver resection may be a technically demanding procedure, with the potential of significant blood loss. Over the past decades, improvements in surgical and perioperative care have contributed to reduce peri-operative mortality after hepatectomy [6,7], which lies around 3% in many cohorts [8,9]. With evidence supporting that parenchymal bleeding and blood transfusions associated with poor outcomes providing the impetus to achieve vascular control during liver resection [10,11,12,13], a classical approach for minimizing surgical blood loss is to interrupt liver blood inflow during parenchymal transection. Liver pedicle clamping, or the Pringle maneuver, was first described by James Hogarth Pringle in the beginning of the 20th century as a way to control hemorrhaging for patients with liver trauma [14]. Many surgeons still use pedicle clamping to control bleeding and reduce liver transection time [15,16,17]. With the development of laparoscopic liver resection, the Pringle maneuver has regained some popularity, given that hemostasis can be more difficult to achieve during laparoscopy than in open surgery [18].

Vascular inflow occlusion comes at the cost of inducing ischemia-reperfusion (IR) injury to the remnant liver parenchyma [19]. Depending on the duration of pedicle clamping and the severity of underlying parenchymal abnormalities [20,21,22], the release of radical oxygen species, microvascular changes and the induction of inflammatory pathways during reperfusion can lead to postoperative liver dysfunction, increased perioperative complications and prolonged recovery [23,24,25]. But beyond early peri-operative morbidity, our group, among others, has contributed in demonstrating that liver IR injury promotes the engraftment and proliferation of circulating tumor cells [26,27,28,29]. This is particularly relevant to the surgical context, where mobilization of the liver can lead to a release of tumor cells, which may encounter a favorable environment for engraftment and growth in the remnant ischemic, regenerative liver [30,31,32,33]. These observations are in line with evidence from the liver transplantation field, where it has been shown that prolonged liver warm ischemia is associated with increased tumor recurrence rates after liver transplantation for HCC [34,35,36]. Other known risk factors for negative adverse oncological outcomes after surgery include severe blood loss, prolonged operating time, positive surgical margin, and postoperative complications [13,37,38,39].

The impact of pedicle clamping on early post-operative morbidity has been studied in various randomized controlled trials [40,41,42,43,44,45]. Surprisingly, these studies did not report long-term outcomes. The current systematic review and meta-analysis aims at synthesizing the available evidence to evaluate whether liver inflow occlusion is associated with a shorter overall survival, or increased rates of tumor recurrence.

## 2. Results

### 2.1. Characteristics of the Included Studies, Intervention Groups, and Critical Appraisal

Ninety-two publications were identified and scrutinized for inclusion. After reviewing abstracts, 21 publications were considered potentially eligible. Finally, after full text assessment, 11 studies [46,47,48,49,50,51,52,53,54,55,56] were included (Figure 1). All studies but one [52] were of retrospective design. One study was considered prospective, as it was a secondary, long-term, nested analysis of the results of two randomized trials comparing intermittent pedicle clamping to no clamping [52]. In this secondary analysis, authors assessed tumor recurrence among other relevant oncological endpoints, years after the original publication. To our knowledge, this is the only long-term re-assessment of the many randomized trials evaluating the impact on liver pedicle clamping.

Eight studies originated from China, one from Hong-Kong, one from Japan and one from Italy (Table 1). Overall, the included studies reported data on 8281 patients. Note that Jiang et al. [47] built a propensity score to allow for baseline imbalance between groups, and through this score, only 202 out of the 396 original patients were matched. Therefore, the total number of patients retained in the current systematic review was 8087.

Among all patients, 6505 (80.4%) were male and 6421 (79.3%) displayed liver cirrhosis. Regarding tumor characteristics, 1459 (18%) had multinodular HCC and 1782 (29.9%) patients underwent a major hepatectomy (Ishizuka et al. [51], Jiang et al. [47] and Li et al. [55] did not report this information in their study, and the last percentage was calculated out of 6127 cases). Regarding the surgical approach, seven studies specified that only cases of open liver resection were included. Famularo et al. [46], Jiang et al. [47], Li et al. [55], and Liu et al. [54], did not provide further details on the surgical technique.

Pedicle clamping was applied in 4674 (57.8%) patients, either continuous or intermittent. There was marked variability in terms of the type and duration of clamping. Nine studies used intermittent clamping [46,47,48,49,50,52,54,55,56]. The two other studies used continuous inflow occlusion [51,53] and stratified their study population in groups undergoing increasing duration of clamping. Characteristics of the included studies are summarized in Table 1 and Table 2.

We designed a network plot to facilitate the review of the study groups and comparisons evaluated in the current systematic review (Figure 2). The most common comparison was between “intermittent pedicle clamping” and “no clamping”.

When comparing, by meta-analysis, the baseline characteristics of the study groups (Appendix A), we found no evidence to support a difference in terms of gender (*p* = 0.961), presence of underlying liver cirrhosis (*p* = 0.139), blood level of alpha-fetoprotein (*p* = 0.558), tumor size (*p* = 0.673), or in the proportion of patients with multinodular HCC (*p* = 0.812). However, a finer qualitative appraisal by the investigators revealed that some studies did in fact carry some baseline imbalance, especially in terms of tumor size and adequacy of the follow-up (Figure 3). On the basis of the Newcastle–Ottawa quality rating assessment, eight out of 11 studies [46,47,48,52,53,54,55,56] were considered of high quality (NOS ≥ 6).

### 2.2. Meta-Analysis of the Impact of Pedicle Clamping on Patient Survival and Tumor Recurrence

Data from seven studies were used in the quantitative synthesis [46,48,49,52,53,54,56]. These studies provided a comparison of one (or more) pedicle clamping group(s) vs. a negative control group that did not undergo any clamping. Other studies could not be pooled, mostly due to the lack of an adequate control intervention [55]. We did not retain the study by Huang et al. [50] for meta-analysis because it compared pedicle clamping with a selective hemi-hepatic inflow occlusion technique, and no other comparator was reported. Similarly, Ishizuka et al. [51] could not be used because there was no control group without pedicle clamping. The study by Jiang et al. [47] was not included, because the occlusion group included patients with both total and partial pedicle clamping, making this group ill-defined. Note that three studies [46,53,54] provided multiple clamping durations and were therefore entered in the analysis as distinct comparisons.

For the primary outcome of interest, meta-analysis indicated that patients who underwent pedicle clamping had a significantly shorter pooled overall survival, as compared to those without clamping (standardized mean difference = −0.17 (95%CI: −0.298 to −0.047), *p* = 0.007, *I*^2^ = 76.8%, Figure 4A). This indicates a small effect size and corresponds to a 4.8 months shorter overall survival (95%CI: −7.6 to −2.0, weighted mean difference). We observed a trend of similar magnitude when looking at disease-free survival, but results were not statistically significant (standardized mean difference = −0.11 (95%CI −0.27 to +0.048), *p* = 0.174, *I*^2^ = 85.7%; Figure 4B). Of note, while we could not incorporate the results of Lee et al. [52] in this analysis on the primary outcome (due to insufficient data to extract median survival), one may object that the results would have shifted the meta-analysis towards the null. Indeed, in this study (55), the no clamping group had a significantly longer survival than the clamping group.

We next assessed whether pedicle clamping was associated with tumor recurrence. To this end, we compared, by random effects meta-analysis, the observed probability of recurrence throughout the assessed literature (Figure 4C). Consistent with our findings on the primary outcome, we found some evidence to support that, as compared to patients who had undergone clamping-free liver resection, those who received pedicle clamping were at risk of tumor relapse (pooled odds ratio 1.36 1.01 to 1.83. *p* = 0.044, *I*^2^ = 50.7%).

### 2.3. Investigation of Heterogeneity with Sensitivity Analyses and Meta-Regression

Because high heterogeneity was observed in our meta-analysis (*I*^2^ > 75%), we did several sensitivity analyses (Table 3) to explore potential sources of heterogeneity. In particular, we looked at the impact of the following factors: (i) duration of pedicle clamping (pooling only studies with prolonged clamping time), (ii) methodological quality of the studies (pooling only studies at low-risk of bias, or NOS ≥ 6), (iii) relevance of the imputation techniques that we used for missing data (by running all meta-analyses once again, but this time assuming a worst-case scenario where missing values were filled with the highest observed variance) and (iv) by including the study by Huang et al. [50] despite the unclear nature of the subgroup entitled “other pedicle clamping”.

Interestingly, clamping time had the largest impact on the statistical heterogeneity. By restricting the analysis to studies (or subgroups of patients) with prolonged clamping time, we could suppress heterogeneity (standardized mean difference = −0.22 (95%CI −0.30 to −0.14), *p* = <0.001 *I*^2^ = 0%, Q Test *p* = 0.436; Figure 5). Other factors did not have a marked impact on statistical heterogeneity (Table 3).

We further evaluated whether methodological quality of the included studies may have affected the results. To this end, we did a meta-regression analysis plotting the methodological quality against the main effect estimate, and found that studies of increasingly robust design tended to identify an increasingly deleterious impact of pedicle clamping on overall survival (Figure 6).

## 3. Discussion

The current systematic review and meta-analysis, which aggregates data on over 8000 patients with resectable HCC, evaluated the impact of hepatic pedicle clamping during liver resection on long-term oncological outcomes. We found that portal vascular clamping was associated with a shorter overall survival and higher tumor recurrence rates. This potentially harmful effect of pedicle clamping was more pronounced in patients undergoing longer clamping times (>15 min). Remarkably, sensitivity analysis where only studies with prolonged clamping times were included, allowed to suppress the heterogeneity observed in the meta-analysis, confirming that clamping time is indeed an important factor affecting survival. A similar trend was identified in terms of disease-free survival, but results did not reach statistical significance.

A recent systematic review was published on a similar subject [57], reducing the originality of the current work. In their meta-analysis, Lin et al. included six studies that only compared intermittent portal clamping to no clamping [41,46,48,49,50,56]. Five of them were pooled quantitatively. Using this approach, authors did not identify a significant difference in terms of either overall or disease-free survival, albeit at the one-year timepoint. Lin et al. [57] concluded that the deleterious effect of inflow occlusion was relevant mostly on short-term outcomes. The main difference with our meta-analysis is that we also included studies comparing continuous liver pedicle clamping to no clamping, hypothesizing that both of these maneuvers provoke. Our analysis also went one step further and also looked at the impact of clamping time.

In 2013, Matsuda et al. looked at the impact of hepatic pedicle clamping on outcome of liver resection for colorectal liver metastasis [58]. Authors did not find significant difference terms of overall or disease-free survival, or intrahepatic recurrence between patients that underwent liver pedicle clamping and those who did not. Such discrepant results may be, at least in part, due to the markedly lower sample size in the review by Matsuda et al. (*n* = 2114 patients), and to the marked differences that exist in the mechanisms and patterns of tumor recurrence between HCC and colorectal liver metastasis.

Hepatic pedicle clamping is still widely used around the world. Results of a 2013 worldwide survey reported that liver pedicle clamping was used routinely in 50% of the surveyed hospitals [16]. In Japan, it is estimated that pedicle clamping is performed routinely in 25% of segmentectomies, in 9% of lobectomies, and in 34% depending on the indication [59]. A recent American national database analysis reported that American surgeons perform pedicle clamping in more than 25% of cases [60]. In Europe, this maneuver is used routinely by 20% of surgeons, and by 71% of them depending on the surgical indication [61]. Importantly, evidence gathered from randomized controlled trials points out that, when performed by expert hands, liver resection can be safely performed without pedicle clamping [41,43]. Similarly, there as to whether pedicle clamping significantly decreases blood loss or reduces operating time [40,42,62].

Our meta-analysis is the first to show that portal triad clamping may have a harmful effect on long-term oncological outcomes. On the basis of this result, and in light of the absence of a clear benefit on short term outcomes, we consider that the use of vascular inflow occlusion during liver resection should be used for selected cases, or as a rescue option, and not as the rule in when planning routine hepatectomy. Selected cases may include redo hepatectomy, large or deeply located tumors [13], and complex laparoscopic cases. With these exceptions in mind, the number of patients requiring pedicle clamping should be limited, by focusing other important components of modern liver surgery, such as maintaining low central venous pressure, choosing the appropriate dissection plane, and cautiously ligating middle-size vessels and biliary canaliculi during parenchymal transection.

Nine of the included studies were published by Chinese research groups (one study was from Japan, and one from Italy), meaning that the results presented in the current meta-analysis may hardly be generalizable to non-Asian populations. Furthermore, there are other limitations to our work. First, with the exception of the study of Lee et al. [52], only retrospective studies were included, putting by definition our results at risk of several types of bias. The main putative bias in the current setting is bias by clinical indication, whereby surgeons may have been inclined to perform pedicle clamping in face of a given surgical situation (nature of the hepatectomy, location and size of the tumor, occurrence of major bleeding), rather than as a planned maneuver. Second, there was significant heterogeneity in our meta-analysis, and this calls for a cautious interpretation of the pooled-effect estimates reported herein. In this regard, we consider that the clinical variability in the included studies may have markedly contributed to statistical heterogeneity, as exemplified by the abrogation of heterogeneity when we pooled studies with similar clamping times. Note that long inclusion periods in some of the studies assessed herein [50,53,54] may also have had an adverse impact on statistical heterogeneity (due to practice changes in the clinical management of older vs. newer cases). Third, to enhance the number of group comparisons, we subdivided five of the included studies [46,48,49,53,54] into subgroups, and analyzed them as separate studies. Finally, while we opted for standardized and validated data imputation techniques for missing standard deviations, this may have affected the accuracy of our effect estimates. Note that the trends identified in the current meta-analysis remained robust even when applying a worst-case scenario with very large standard deviation.

Our results demonstrate that hepatic blood flow occlusion, during liver surgery for HCC resection, seems to negatively impact patient survival. This is in line with other pieces of evidence showing that an injured liver parenchyma portends an elevated risk of tumor recurrence [34,35,36]. More evidence from prospective studies is necessary on this topic and an ideal way to fill this gap would be to perform an individual patient meta-analysis, collecting the long-term outcomes from the numerous randomized controlled trials available on this subject, but focused on peri-operative outcomes.

## 4. Methods

### 4.1. Literature Search, Study Selection, and Outcomes of Interest

The protocol for this systematic review was registered in the Prospero database (CRD42018102641). We formulated a structured keyword search in Medline/PubMed, from January 1960 until 6 April 2020, in order to identify studies evaluating the impact of liver pedicle clamping on long-term outcomes after liver resection for HCC. The query was as follows:
***[**(portal) and (clamping)**] OR [**(inflow) and (occlusion)**] OR [**(hepatic) and (inflow)**] OR [**(liver [MeSH Terms) and (clamping)**] OR [**(pedicle) and (clamping)**] OR [**(pringle) and (maneuver)**]******AND******[**(hepatocellular carcinoma [MeSH Terms]) and (recurrence)] **OR** [(survival)] **OR** [(long-term)] **OR** [(prognosis)]*

Studies had either to provide at least a comparison between a liver pedicle-clamping group to a no clamping group in a head-to-head fashion. Studies comparing distinct forms of pedicle clamping were also retained. We only included studies that reported an estimation of patient overall survival beyond a 12-months period post-liver resection. Given that numerous randomized trials evaluated the impact of pedicle clamping on early post-operative morbidity [40,41,42,43,44,45], we particularly looked for secondary, long-term analyses of these studies. We excluded patients undergoing liver resection for colorectal liver metastases (or other indication than HCC) and case-series with less than ten patients. For the purpose of this review, we only assessed studies written in English.

### 4.2. Data Extraction

Two independents investigators (CH.W. and B.M.) scrutinized the database search, evaluated potential articles for inclusion, assessed study quality, and extracted data according to a pre-established review form (available upon request). Discrepancy was resolved by reaching a consensus with the senior reviewer (L.A.O.). We extracted the following data from the individual studies: author name, date of publication, country where the study took place, epidemiological design, the period during which the study took place, the total number of patients included and their characteristics (gender, age, presence of cirrhosis, number and size of the tumor nodules). We retrieved information on the technical aspect of the pedicle clamping method (continuous vs. intermittent, tourniquet vs. direct clamp application), the duration of liver inflow occlusion, operative time, blood loss, and the extension of liver resection (major resection being considered when ≥3 segments were resected). Next, we retrieved long-term outcomes, collecting informations on all of the following endpoints: overall survival (primary outcome of interest), disease-free survival, point estimates of survival probability (1, 3, 5-years mean survival time), rate of tumor recurrence and duration of follow up.

### 4.3. Risk of Bias Assessment

Investigators assessed the risk of bias using the Newcastle–Ottawa scale (NOS). Briefly, each of the included study was evaluated on (i) the selection of the patients in each group (four items), (ii) the comparability between study groups (two items) and (iii) the method to assess outcomes (three items) [63]. Studies may be rated from 0 to 9, with 9 indicating very low risk of bias. Of note, the number of tumor nodules and size of the largest tumor were chosen as factors to evaluate the comparability of the study groups.

### 4.4. Quantitative Synthesis and Statistical Analysis

We anticipated that a variety of pedicle clamping techniques would be reported throughout the literature. Therefore, we classified the distinct comparisons made in the individual studies by constructing a network plot, as described by Chaimani et al. [64]. In such a plot, the size of each dot represents the number of study arms, and the thickness of each connecting line represents the number of comparisons made between the given groups (e.g., whole pedicle clamping vs. no clamping, intermittent clamping vs. continuous clamping, prolonged vs. short continuous clamping). Whenever data were sufficient, outcomes were pooled and compared by random effects meta-analysis [65]. The primary comparison of interest to this meta-analysis looked at (a) patients undergoing portal triad clamping (either intermittent or continuous) *versus* (b) those undergoing no clamping at all. In this main analysis, studies reporting more than one comparison of clamping type or duration were considered as separate studies [66]. The aggregated effect size was expressed as a standardized mean difference in survival.

Between-study heterogeneity was calculated using the *I^2^* statistic, and was explored using several approaches. First, sensitivity analyses were performed to address the putative sources of heterogeneity. As another approach, we conducted meta-regression analyses to test for an association between bibliometric and clinical characteristics of the individual studies and their respective effect estimates. Factors assessed by meta-regression included study quality, the impact factor of the journal where studies were published, the extent of the liver resection, the number of tumor nodules, the size of the largest tumor, the median age of the cohort.

In case of missing summary statistics (such as mean values and standard deviations), we estimated them from medians and percentiles as proposed elsewhere [66,67]. Moreover, when necessary, relevant data were obtained by digitalizing results from the original figures, via high magnification and point estimation with the software GetData Graph Digitizer [68]. Statistical analyses were done using Stata software (v.15, College Station, TX, USA).

## 5. Conclusions

In conclusion, the current systematic review and meta-analysis suggests that prolonged liver pedicle clamping may be associated with shorter survival after liver resection for hepatocellular carcinoma. The risks and benefits of liver pedicle clamping need to be carefully weighed for each patient, taking into account local anatomy and aiming to optimize surgery and anesthesia in order to achieve minimal blood loss during parenchymotomy.

## Figures and Tables

**Figure 1 cancers-13-00637-f001:**
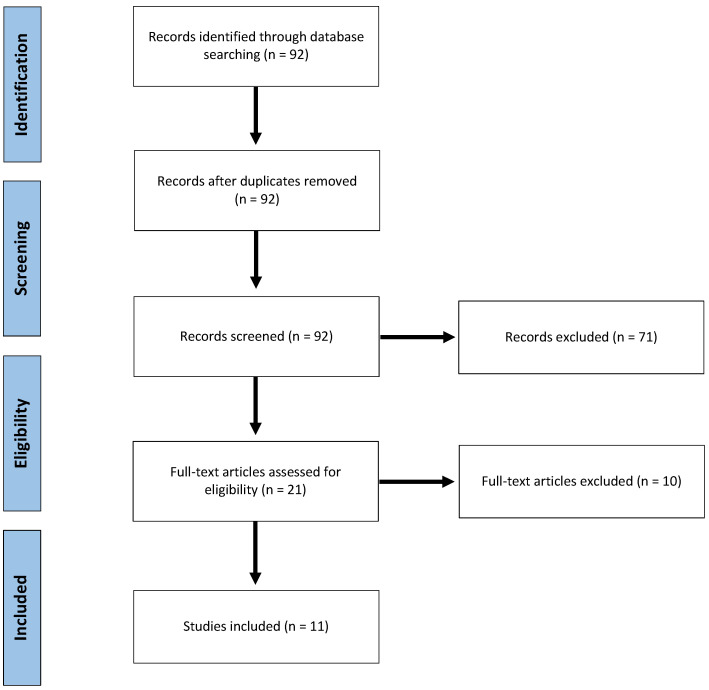
Flow Chart of the inclusion-exclusion process, according to the Preferred Reporting Items for Systematic Reviews and Meta-Analyses.

**Figure 2 cancers-13-00637-f002:**
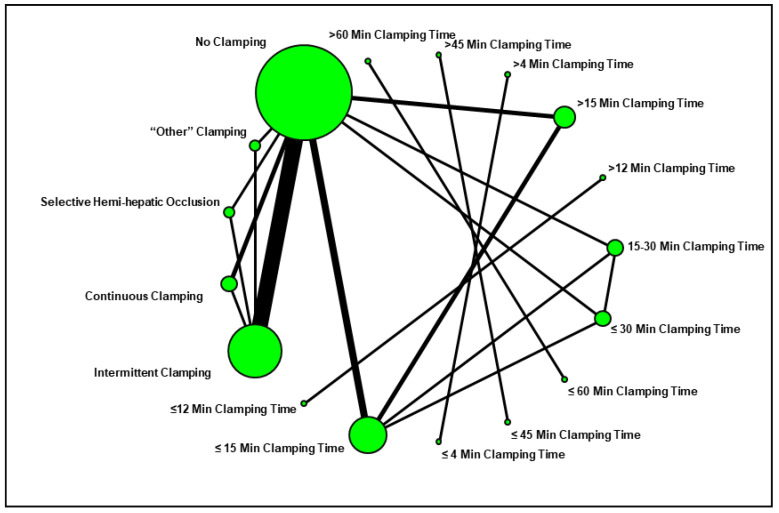
Network plot summarizing the comparisons assessed in the current systematic review.

**Figure 3 cancers-13-00637-f003:**
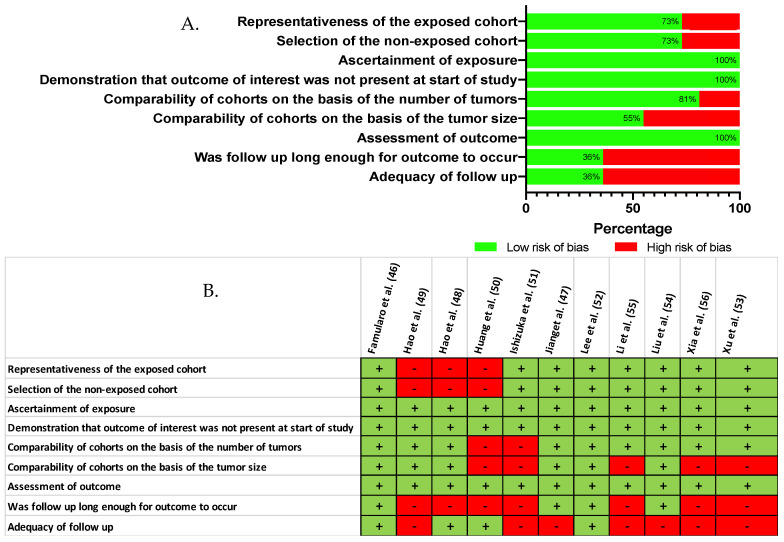
Critical appraisal of the reviewed literature, according to the Newcastle Ottawa Scale (**A**) Reviewer’s judgment about the risk of bias item presented as percentages across all included studies (NOS) (**B**) Risk of bias as estimated by the authors for each item and each study that was included [46,47,48,49,50,51,52,53,54,55,56].

**Figure 4 cancers-13-00637-f004:**
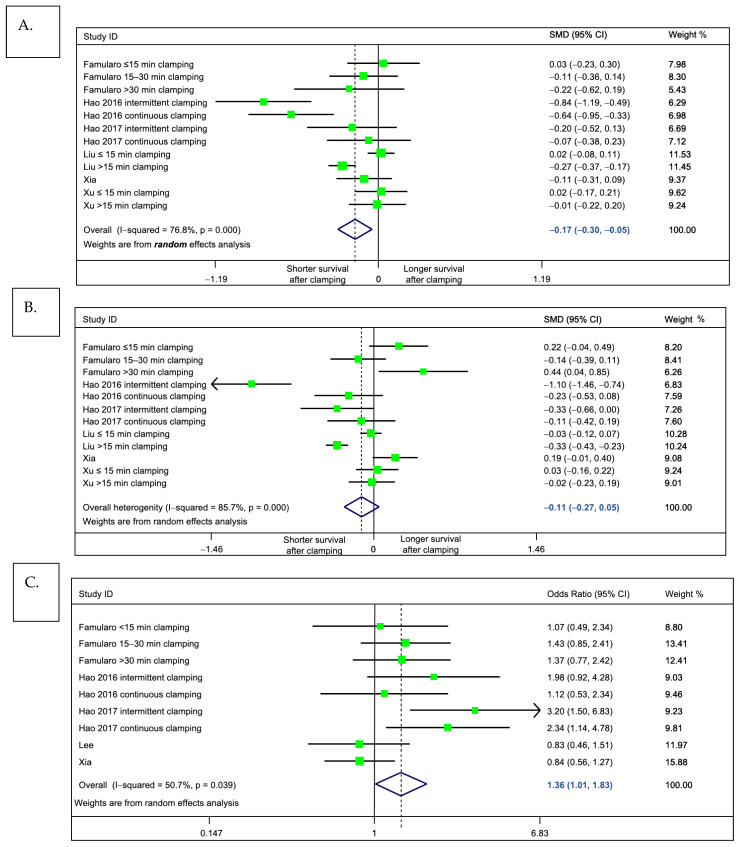
Meta-analysis comparing outcomes of patients undergoing liver pedicle clamping, versus no-clamping. Forest plots depict results obtained by random-effects meta-analysis. The vertical solid line depicts the null hypothesis, and the dashed line indicates the pooled effect estimate. (**A**) Primary outcome, overall survival (standardized mean difference). (**B**) Secondary outcome, disease-free survival (standardized mean difference). (**C**) Secondary outcome, overall probability of tumor recurrence (odds ratio). Note that in the main meta-analyses, the study by Huang et al. was excluded, because it included a subgroup of patients with unclear pedicle clamping technique (“other clamping”). In a sensitivity analysis including this study, the pooled effect estimate did not differ from the main analysis.

**Figure 5 cancers-13-00637-f005:**
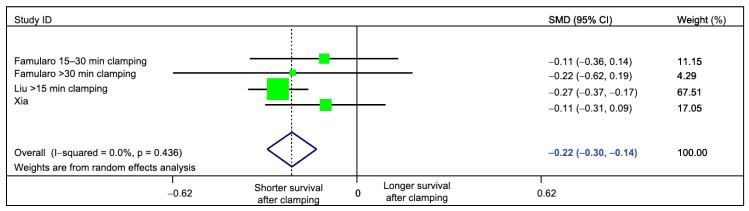
Sensitivity analysis, evaluating the impact of clamping time on the measured statistical heterogeneity. Only studies comparing prolonged (≥15 min) clamping time versus no-clamping were pooled. Residual heterogeneity was 0%.

**Figure 6 cancers-13-00637-f006:**
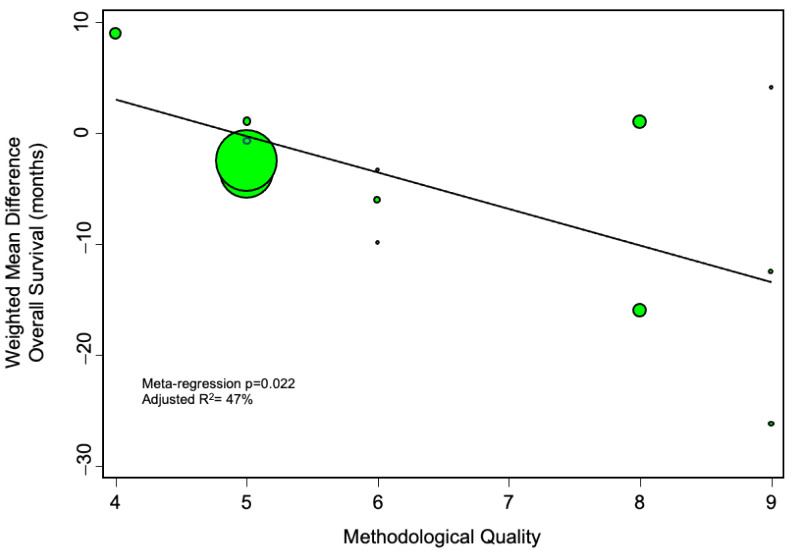
Meta-regression analysis, evaluating the impact of methodological quality (as assessed by the Newcastle-Ottawa scale) on the calculated pooled estimate. The more negative the weighted mean difference, the more harmful the effect of pedicle clamping on overall survival. The diameter of the individual studies indicates their relative weight in the meta-analysis.

**Table 1 cancers-13-00637-t001:** Study characteristics.

Study	Year	Country	Design (Period)	Primary Outcome	Secondary Outcome	Total No. of Patients	Age, Year (Range/SD)	Follow Up (Months)
Famularo et al. [46]	2018	Italy	RCS (2001–2015)	OS	DFS	441	67 (58.7–73.0) ^a^	60
Hao et al. [49]	2016	China	RCS (2010–2012)	OS	DFS	266	48, 5 ± 10.3 ^b^	25
Hao et al. [48]	2017	China	RCS (2007–2010)	OS	DFS	355	48, 5 ± 10.3 ^b^	49.7 (6–66) ^a^
Huang et al. [50]	2014	China	RCS (1998–2008)	OS	DFS	1549	56, 1	68.4 ± 57.8 ^b^
Ishizuka et al. [51]	2011	Japan	RCS (2000–2008)	OS	DFS	357	67 (28–85)^a^	32.7 ± 23.6 ^b^
Jiang et al. [47]	2017	China	RCS (2004–2009)	OS	DFS	202	47	60.7
Lee et al. [52]	2019	Hong Kong	nested-analysis of two randomized trials (2008–2011, 2013–2016)	OS	DFS	176	59.3 (27–84)	44.3 (0.2–120.2) ^a^
Li et al. [55]	2016	China	RCS (2003–2006)	OS	DFS	1401	NR	120
Liu et al. [54]	2016	China	RCS (1999–2008)	OS	DFS	2368	NR	52.2
Xia et al. [56]	2013	China	RCS (2001–2006)	OS	DFS	386	53	120
Xu et al. [53]	2017	China	RCS (1989–2011)	OS	DFS	586	56, 1	36 (1–157) ^a^

RCS = retrospective cohort study, OS = overall survival, DFS = disease-free survival, ^a^ = data presented as median and range, ^b^ = data presented as mean +/− SD = standard deviation.

**Table 2 cancers-13-00637-t002:** Patient, tumor and treatment characteristics according to clamping groups.

														5-Year OS		5-Year DFS	
Study	Group	No. of Patients	Age, Year	Male, n (%)	Cirrhosis, n (%)	Clamping Interval (min)	Total Clamping Time, min	AFP (ng/mL)	Multiple Lesions, n (%)	Tumor Size, cm	Surgery Duration, min	Major Resection, n (%)	Blood Loss, mL	Rate (%)	Median, Months	Rate (%)	Median, Months
Famularo et al. 2018 [46]	IPC ≤ 15 min	176	65.1 (58.2–72) ^a^	145 (82.4)	144 (82.3)	15; 5	23 (14–30) ^a^	25.4 (7.3–159.7) ^b^	42 (24)	56 (31.8) ^e^	170 (140–220) ^a^	22 (12.5)	400 (50–700) ^a^	/	60.6 (24.9–96.8) ^g^	/	26.7 (15.7–37.7) ^g^
	16–30min														44.0 (33.0–54.9) ^g^		
	>30 min														30.3 (23.9–36.8) ^g^		
	NPC	265	67.6 (59.2–73.9) ^a^	199 (75.1)	214 (80.8)	NA	NA	21.8 (7.3–214.2) ^b^	54 (20.5)	65 (24.5) ^e^	70 (125–210) ^a^	36 (12.2)	300 (50–600) ^a^	/	56.5 (37.1–75.9) ^g^	/	24.9 (18.1–31.7) ^g^
Hao et al. 2016 [49]	IPC	78	51.7 ^b^	60 (77.9)	78 (100)	15; 5	31.5 ± 12.5 ^b^	/	31 (39.7)	46 (59.0) ^f^	187 ± 31 ^b^	50 (64.1)	396 ± 78 ^b^	72.7^g^	19 ± 4.2 ^b, h^	63.6 ^h^	14.2 ± 4.6 ^b, h^
	CPC	128	53.6 ^b^	101 (78.9)	128 (100)	NA	27.7 ± 7.3 ^b^	/	45 (35.2)	70 (54.7) ^f^	132 ± 26 ^b^	77 (60.2)	422 ± 75 ^b^	79.9^g^	20 ± 3.8 ^b, h^	75.8 ^h^	18 ± 4.8 ^b, h^
	NPC	60	55.0 ^b^	48 (80.0)	60 (100)	NA	NA	/	21 (35.0)	34 (56.7) ^f^	140 ± 22 ^b^	15 (25.0)	405 ± 83 ^b^	83.1^g^	22.5 ± 4.1 ^b, h^	78 ^h^	19.0 ± 4.1 ^b, h^
Hao et al. 2017 [48]	IPC	113	51.7 ^b^	76 (67.3)	113 (100)	15; 5	/	/	46 (40.7)	73 (35.4) ^f^	/	71 (62.8)	/	44.9	46.3	42.5	39.4
	OPC	190	53.6 ^b^	130 (68.4)	190 (100)	NA	/	/	73 (38.4)	114 (40) ^f^	/	110 (57.9)	/	58	52.9	50.9	47.3
	NPC	52	55.0 ^b^	37 (71.1)	52 (100)	NA	NA	/	18 (34.6)	30 (42.3) ^f^	/	27 (51.9)	/	57.7	56.2	49.6	51.4
Huang et al. 2014 [50]	IPC	712	56.1 ± 16.5 ^b^	505 (70.9)	518 (72.8)	15; 5	47.4 ± 38.7 ^b^	8176.3 ± 3211.5 ^b^	199 (27.9)	8.6 ± 7.8 ^b^	172.1 ± 95.9 ^b^	338 (47.5)	1146.3 ± 895.2 ^b^	42	/	22	/
	SPC	219	57.2 ± 19.4 ^b^	162 (74.0)	164 (74.9)	30; 5	53.1 ± 33.5 ^b^	6776.3 ± 2771.8 ^b^	64 (29.2)	6.3 ± 4.4 ^b^	200.4 ± 119.4 ^b^	78 (35.6)	1311.8 ± 785.4 ^b^		/		/
	NPC	618	54.2 ± 22.1 ^b^	473 (76.5)	322 (52.1)	NA	NA	6421.2 ± 5641.9 ^b^	185 (29.9)	7.7 ± 5.1 ^b^	248.8 ± 146.1 ^b^	289 (46.8)	1428.6 ± 1123.7 ^b^	35	/	18	/
Ishizuka et al. 2011 [51]	PC ≤ 60min	242	68 (28–85) ^a^	194 (80.2)	126 (52.1)	15; 5	53.7 ± 26.1 ^b^	24 (3–779 000) ^a^	54 (22.3)	2.5 (0.5–25) ^a^	280 (123–683) ^a^	/	437 (60–6840) ^a^		32.1 ± 23.7 ^b^		/
	>60min	115	67 (36–84) ^a^	91 (79.1)	56 (48.7)	15; 5		60 (2–670 000) ^a^	37 (32.2)	3.7 (0.6–17) ^a^	348 (191–875) ^a^	/	631 (50–7240) ^a^		32.6 ± 22.5 ^b^		/
Jiang et al. 2017 [47]	IPC	101	48.3 ± 11.1 ^b^	91 (90.1)	73 (72.3)	20; 5	/	34 (33.4) ^i^	10 (9.9)	5.7 ± 2.8 ^b^	/	/	300 (200–500) ^a^	39.6	/	14.9	/
	NPC	101	46.8 ± 12.0 ^b^	88 (87.1)	70 (69.3)	NA	NA	36 (35.6) ^i^	12 (11.9)	5.6 ± 2.7 ^b^	/	/	300 (200–500) ^a^	45.5	/	14.2	/
Lee et al. 2019 [52]	IPC	88	58 (38.0–84.0)	75 (85.2)	53 (60.2)	15; 5	45.0 (15–87) ^a^	28 (1–191 009) ^a^	24 (27.3)	3.9 (1.0–12.3) ^a^	230 (120–560) ^a^	37 (42.0)	331.5 (50–3600) ^a^	72.1	/	49.7	/
	NPC	88	60.5 (27.0–81.0)	75 (85.2)	50 (56.8)	NA	NA	25 (1–16 246) ^a^	20 (22.7)	3.5 (1.0–18.0) ^a^	230 (110–480) ^a^	34 (38.6)	310.0 (50–3160) ^a^	58.1	/	40.3	/
Li et al. 2016 [55]	IPC > 4min	408	190 (46.6) ^c^	356 (87.3)	336 (82.4)	15; 5	NA	277 (67.9) ^j^	58 (14.2)	244 (59.8) ^f^	/	/	/	53.8	/	47	/
	PC ≤ 4min or NPC	993	446 (44.9) ^c^	843 (84.9)	871 (87.7)	NA	NA	649 (65.4) ^j^	142 (14.3)	370 (37.3) ^f^	/	/	/	53.9	/	47.2	/
Liu et al. 2016 [54]	IPC ≤ 15min	866	206 (23.8) ^d^	753 (87.0)	757 (87.4)	15; 5	17.5 ± 7.2 ^b^	568 (65.6) ^j^	90 (10.4)	8.2 ± 4 ^b^	/	90 (10.4)	235 ± 110 ^b^	/	61	/	51
	>15 min	724	175 (24.2) ^d^	564 (77.9)	621 (85.8)	15; 5		481 (66.6) ^j^	115 (15.9)	8.5 ± 4 ^b^	/	115 (15.9)	231 ± 105 ^b^		44		40
	NPC	778	189 (24.3) ^d^	649 (83.4)	700 (90.0)	NA	NA	528 (67.9) ^j^	101 (13.0)	8.1 ± 4 ^b^	/	63 (8.1)	377 ± 161 ^b^		60		52
Xia et al. 2013 [56]	IPC	224	48 (21–78) ^a^	173 (77.2)	169 (75.4)	15; 5	50 (30–98) ^a^	143 (63.8) ^i^	79 (35.3)	6.4 (2.8–20.2) ^a^	/	93 (41.5)	500 (50–3600) ^a^	43.9	/	17.9	/
	NPC	162	57 (18–79) ^a^	119 (73.5)	128 (79.0)	NA	NA	93 (57.4) ^i^	40 (24.7)	5.9 (2.9–21.3) ^a^	/	77 (47.5)	450 (50–3800) ^a^	45.1	/	13.8	/
Xu et al. 2017 [53]	CPC < 15min	163	56.18 ± 11.77 ^b^	139 (85.3)	114 (69.9)	NA	11.92 ± 3.78 ^b^	2725.79 ± 9423.36 ^b^	28 (17.2)	4.46 ± 3.06 ^b^	197.01 ± 60.58 ^b^	55 (33.7)	405.21 ± 924.4 ^b^	45.7	/	35	/
	≥15min	127	55.87 ± 11.15 ^b^	113 (89.0)	93 (73.2)	NA	25.05 ± 3.60 ^b^	8736.39 ± 39,952.35 ^b^	24 (18.9)	6.28 ± 3.26 ^b^	227.40 ± 55.14 ^b^	50 (39.4)	618.50 ± 606.2 ^b^	42.5	/	33.2	/
	NPC	296	56.10 ± 12.05 ^b^	246 (83.1)	221 (74.7)	NA	NA	4526.09 ± 15,714.95 ^b^	47 (15.9)	5.76 ± 4.12 ^b^	215.69 ± 88.58 ^b^	126 (42.6)	783.55 ± 1554.4 ^b^	39.7	/	33.9	/

PC = portal clamping, IPC = intermittent pedicle clamping, NPC = no pedicle clamping, CPC = continuous pedicle clamping, OPC = other pedicle clamping types (continuous, pre-conditioning, selective PC), SPC= selective pedicle clamping, NA = not available, OS = overall survival, DFS = disease free survival, AFP = alpha-fetoprotein. ^a^ data presented as median and range, ^b^ data presented as mean +/- SD (standard deviation), ^c^ data presented as number of patient (%) ≤ 50 years, ^d^ data presented as number of patient (%) > 60 years, ^e^ and ^f^ data presented as number of patient (%) with tumor size ≥ 5 cm or >5 cm, respectively, ^g^ data presented as median (95% CI), ^h^ results at 1 year, ^i^ data presented as number of patient (%) with AFP ≥ 400ng/mL, ^j^ data presented as number of patient (%) with AFP ≥ 20ng/mL.

**Table 3 cancers-13-00637-t003:** Results of the sensitivity analyses.

Factor Assessed	Number of Comparisons	Pooled Estimate (95% CI)	*p*-Value	Heterogeneity
Prolonged (>15 min) clamping time	4	SMD = −0.220 (−0.304 to −0.137)	<0.001	Q test *p* = 0.436, I^2^ = 0%
Standard deviation assumption using a worst-case scenario	11	SMD = −0.150 (−0.272 to −0.028)	0.016	Q test *p <* 0.1, I^2^ = 73%
Pooling only high-quality studies only (NOS ≥ 6)	8	SMD = −0.110 (−0.223 to −0.002)	0.055	Q test *p <* 0.1, I^2^ = 60%
Inclusion of the study by Huang et al. [50]	12	SMD = −0.157 (−0.294 to −0.019)	0.026	Q test *p <* 0.1, I^2^ = 84%

SMD = Standardized Median Difference.

## Data Availability

The data presented in this study are available in this article (and Appendix A).

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
