# Peer review of "Shorter Survival after Liver Pedicle Clamping in Patients Undergoing Liver Resection for Hepatocellular Carcinoma Revealed by a Systematic Review and Meta-Analysis"

_cancers, 2021, doi:10.3390/cancers13040637_

Round 1

Reviewer 1 Report

I have reviewed the revised manuscript. And I felt it very informative article for Cancers. I agree with acceptance of this article.

Reviewer 2 Report

The authors responded to my doubts and questions posed by myself following reading of the first version of the paper. The other data or corrections have been introduced everywhere where they were required.

This manuscript is a resubmission of an earlier submission. The following is a list of the peer review reports and author responses from that submission.

Round 1

Reviewer 1 Report

The authors have investigated the relationships between hilar pedicle clamping and prognosis of HCC after liver resection. It is very surprising results for readers, because Pringle's maneuver is one of the most important technique of liver resection. However, there are some points to be revised for acceptance.

Major

  1. Were these cases of enrolled studies including laparoscopic liver resections? If yes, these backgrounds should be also presented and evaluated for DFS, and OS.
  2. Variations of blood loss and the rate of anatomical resection, and the interval of enrolled patients seems to be so long. Are there any possibility of fetters on negative results by old cases?
  3. Usual oncological negative factors in surgery are blood loss, operating time, surgical margin or exposure of tumor cells, and postoperative complications. Please mention about these factors.

Reviewer 2 Report

The aim of this meta-analysis was to evaluate if pedicle clamping is associated with tumor recurrence and shorter survival in hepatocellular carcinoma (HCC) patients.

The aims and hypotheses of the work are clear, scientifically justified and do not raise any objections. The paper is very important from clinical point of view and may carry practical importance. The meta-analysis with data on over 8000 patients with resectable HCC suggests that prolonged liver pedicle clamping may be associated with shorter survival after liver resection for HCC. According to the observations of the authors, the risks and benefits of liver pedicle clamping need to be carefully weighed for each patient. The results of the work are very interesting, presented correctly, legibly and do not raise any objections. The most interesting results and work limits are also clearly stated. The aim of the work has been achieved. The discussion is written perfectly, all results and doubts are explained comprehensively. The authors show the need for optimize surgery and anesthesia in order to achieve minimal blood loss during liver resection. I have no comments on the description of the methodology (e.g., literature search, study selection, data extraction, risk of bias assessment). Appropriate statistical analyzes were used for the research.  

Minor editorial notes:

Line 84 - Please correct the numer of records identified through database searching –you wrote ninty-three publications, on Figure 1 is n=92;Line 113-118 - please complete the legend with an explanation of some abbreviations, e.g. NA;

Line 129 – instead of “alpha-feto protein” should be better “alpha-fetoprotein”;

Line 218 – instead of “>15 mins.” should be “>15 min.” like in all text;

Line 348 - the last sentence in description of statistical analysis used is a little unclear; should be “were done with (using) Stata software“. Is it true that the authors utilized Stata Statistical Software?The References list - The DOI number is missing for all references. Please complete this and check other details of references as required by the journal.

Reviewer 3 Report

By performing a meta-analysis on 11 trials including > 8000 patients, Wassmer et al. demonstrate that liver pedicle clamping may have a deleterious impact on long-term outcomes in patients resected for HCC. The finding is clinically important since the technique is routinely applied in liver surgery and therefore might be translated into clinical patients´ care.

The statistical analyses were thorough and the authors' claims are supported by the data. The obvious limitations of the analysis are transparently stated and discussed by the authors; in particular, the quality of the studies analysed. Moreover, a recent similar analysis is cited and discussed. Likewise, the transferability to non-Asian collectives is rightly questioned.

A few comments:

- Is there information about the pretreatment of the patients and does it influence the outcome?

- since OS was analyzed, are data on further lines of therapy available?
